# Spatio-Temporal Analysis of Ecological Vulnerability and Driving Factor Analysis in the Dongjiang River Basin, China, in the Recent 20 Years

Jiao Wu [1] , Zhijun Zhang [2,3], Qinjie He [1] and Guorui Ma [1,*]

1   State Key Laboratory of Information Engineering in Surveying, Mapping and Remote Sensing,
    Wuhan University, Wuhan 430079, China; wujiaors@whu.edu.cn (J.W.); 2018206190073@whu.edu.cn (Q.H.)
2   Xining Center of Natural Resources Comprehensive Survey, China Geological Survey, Xining 810000, China;
    zhangzhijun@mail.cgs.gov.cn
3   Ministry of Education Key Laboratory of Geological Survey and Evaluation, China University of
    Geosciences (Wuhan), Wuhan 430074, China
*   Correspondence: mgr@whu.edu.cn

**Abstract:** The global ecological environment faces many challenges. Landsat thematic mapper time-series, digital elevation models, meteorology, soil types, net primary production data, socio-economic data, and auxiliary data were collected in order to construct a comprehensive evaluation system for ecological vulnerability (EV) using multi-source remote sensing data. EV was divided into five vulnerability levels: potential I, slight II, mild III, moderate IV, and severe V. Then, we analyzed and explored the spatio-temporal patterns and driving mechanisms of EV in the region over the past 20 years. Our research results showed that, from 2001 to 2019, the DRB was generally characterized as being in the severe vulnerability class, with higher upstream and downstream EV classes and a certain amount of reduction in the midstream EV classes. Moreover, EV in the DRB continues to decrease. The spatio-temporal EV patterns in the DRB were significantly influenced by the relative humidity, average annual temperature, and vegetation cover over the past 20 years. Our work can provide a basis for decision-making and technical support for ecosystem protection, ecological restoration, and ecological management in the DRB.

**Keywords:** ecological vulnerability; driving mechanisms; remote sensing; Dongjiang River Basin





## 1. Introduction

Since the middle of the 20th century, the intensification of human activities has led to frequent climate-change-related disasters [1–4] and the intensification of ecological and environmental crises [5]. These scenarios pose serious challenges to the ecosystems on which humans depend for their survival. Among them, the issue of ecological vulnerability (EV) is particularly prominent. EV was first introduced into ecological theory by Clements as an "ecological staggering zone" [6,7] and was further discussed at the seventh SCOPE (Scientific Committee of Environmental Problems) Conference in 1989 [6,8,9]. The ability of a system to resist external environmental change and to be disturbed and recover itself is defined as EV [10–13]. EV is determined by a combination of internal and external vulnerability [14,15]. Internal vulnerability usually stems from the structure of the ecosystem itself and is mainly influenced by natural conditions such as topography and climate. External vulnerability is influenced by human activities [16]. Exploring regional EV is important for ecological change and socio-economic development.

Scholars around the world have conducted relevant studies on the spatio–temporal evolution patterns of EV and its driving mechanisms [2,13,17–23]. Most of these EV studies are focused on China [21] and can be based on individual cities or regions [24], e.g., on the northwestern part of the Songnun Plain [17], the Tibetan Plateau [18], the Loess Plateau [19], the southwestern karst mountains [10], agricultural and pastoral areas [22], coal mining



areas [2], the southern Shaanxi region [25], islands [26], coastal wetlands [27]; there are also studies that focus on the Bangladesh–China–India–Myanmar economic corridor [21]. Recently, the ecology of red loam hilly areas in southern China has become a focus of attention for many scholars [28].

Dongjiang is the third largest water system in the Pearl River basin and belongs to the South China basin, which is characterized as having a typical tropical–subtropical climate. The Dongjiang River Basin (DRB) is located in the red-loamy hilly region of southern China, which, as a pioneering region for reform and opening up, has seen a rapid economic development over the past 40 years. China's 13th Five-Year Plan supports the construction of an open and innovative transformation of the Pearl River Delta region, namely the Guangdong–Hong Kong–Macao Greater Bay Area. Meanwhile, the DRB is an important drinking water source for the Pearl River Delta and Hong Kong and an essential focus of the sustainable development strategy in the Pearl River Delta region. The socio-economic development of the basin and changes in the natural environment affect the quality of the ecological environment and the sustainable use of resources in the basin [29]. With accelerated urbanization, dramatic climate change, and increased disturbance from human activities, the quality of the ecological environment in the DRB faces unprecedented threats, and thus the DRB has become an ideal area for analyzing EV. Hu [24] used the AHP method to evaluate the EV of Weifang city, China; however, the study used limited the quantitative data and was heavily dependent on qualitative inputs. These factors led to unconvincing results, especially when there were many factors, and it was difficult to precisely determine the weights. Wu [30] used the fuzzy hierarchical analysis method to evaluate the EV of the Yellow River Delta; however, there were drawbacks related to the subjective factor weights and large computational effort. Principal component analysis (PCA) is one of the most commonly used methods for model evaluation, as it can reduce data complexity, identify the most important multiple features, and save significant computational resources, and is easily implemented on a computer. In this paper, PCA was used to model and analyze the time-series EV in the DRB.

Factors commonly used in EV assessment can be divided into two categories: those reflecting human activities, such as population density, GDP per capita, etc.; and ecological and natural conditions, such as slope, temperature, precipitation, vegetative cover, etc. As a result of the complexity of human–nature interactions, there is currently no uniform standard for the selection of factors for EV assessment models [31]. Moreover, there is the problem of regional adaptability in EV assessment models [22]. The ecological environment in the red soil hilly areas of southern China is fragile [32]; furthermore, there is a lack of an EV zoning system. In addition, many of the current studies on EV are analyzed at a regional scale and are less specific to county EV evaluations. Therefore, there is a need to develop a reliable methodology to assess EV in the DRB.

Our work established a dynamic evaluation scheme for monitoring the spatio-temporal EV changes in the DRB and calculated the dynamic EV weights in 2001–2019 using PCA. Finally, we analyzed the spatio-temporal evolution of EV in the DRB and its driving mechanisms over the past 20 years. The research results and methods provide a theoretical basis and technical support for the protection and restoration of the environment within the DRB. The flowchart for this work is shown in Figure 1.

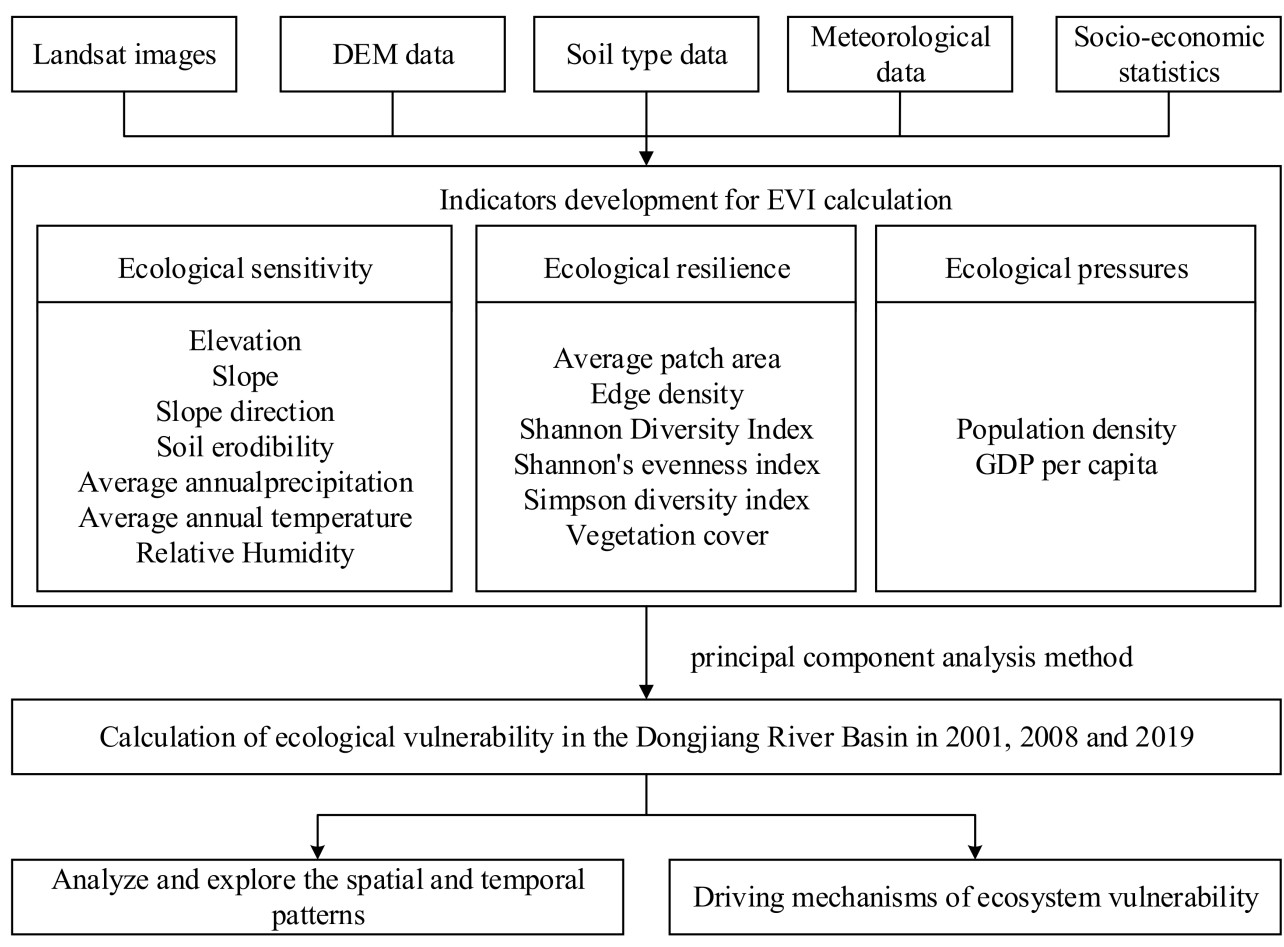

**Figure 1.** Stepwise procedure for spatio-temporal EVI modeling.

## 2. Materials and Methods

### 2.1. Study Area

The DRB (Figure 2) is connected to Meishan in eastern Guangdong; to its west, it is bordered by Shaoguan and Qingyuan in northern Guangdong; the South China Sea and Hong Kong lie to the south, and the Dongjiang source area of southern Ganzhou lies to the north. The geographical location of the DRB is 113°52′–115°52′ E and 22°38′–25°14′ N, with hilly mountains in the center and north. The DRB is a valuable water resource for social development within the basin, providing water to Hong Kong for production, living, and ecology, and to more than 40 million people who live in the urban clusters along the basin. The middle and lower reaches of the DRB are economically developed, and the total GDP of the regions in Guangdong, China, that are supported by Dongjiang water was RMB 4.17 trillion in 2019, accounting for approximately 39% of the total GDP of Guangdong Province.

### 2.2. Evaluation Factors and Data Sources

We studied the "China Guangdong Environmental Protection Department on the Comprehensive Water Environment Improvement Programme of the Dongjiang River Basin (2015–2020)", combined with related studies [33–35], and considered the availability of relevant assessment factors. We selected factors from topography, climate, landscape, vegetation, and socio-economic aspects to construct an EV evaluation system. The data sources are shown in Table 1. There were some differences as regards the data sources and spatial accuracies of the factors. In the Arc GIS10.1 software platform, the WGS 1984 coordinate system and Mercator projection were uniformly used to ensure acceptable spatial coincidences for the factors. Moreover, all of these were unified to a raster image

element size of 30 × 30 m using the ArcGIS10.2 platform to accommodate subsequent spatial calculations. The data sources are shown in Table 1.

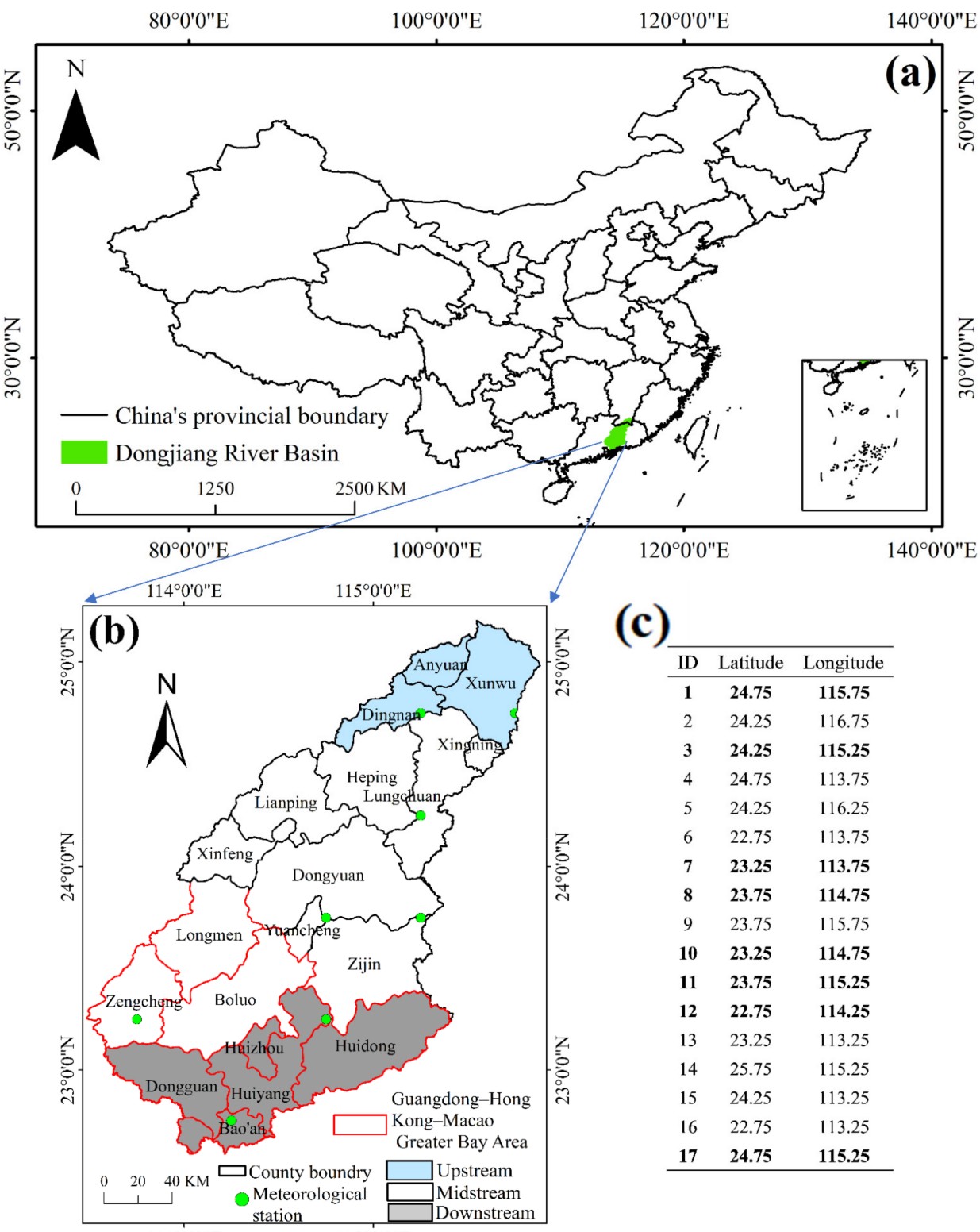

**Figure 2.** Study area. (**a**) Location of the study area in China; (**b**) The extent of the DRB; (**c**) The geographical location of meteorological stations. (The meteorological stations marked in bold are located within the DBR, and the meteorological stations not marked in bold are located outside the DBR).

**Table 1.** Data sources and characteristics.

| Name of Data | Data Production Unit | Data Source Website | Resolution | Processing Method |
|---|---|---|---|---|
| Landsat remote sensing satellite data | USGS | https://earthexplorer.usgs.gov/ | 30 m | Spatial analysis |
| GDEMV2 Elevation Data | Geospatial Data Cloud | http://www.gscloud.cn/ | 30 m | Spatial analysis |
| GDP per capita | Statistical Yearbook of Jiangxi and Guangdong Province, China | – | – | Statistical analysis |
| 1:4 million Chinese soil type data | National Earth System Science Data Sharing Platform | http://www.geodata.cn/ | – | Spatial analysis |
| Population density | WorldPOP dataset | https://www.worldpop.org/ | 100 m | Spatial analysis |
| Meteorological data | China Weather Data website | http://data.cma.cn/ | – | Spatial analysis |

Note: We last accessed the link above on 16 November 2020.

Topographic factors included elevation, slope, slope orientation, soil erosion, and other factors. The GDEMV2 elevation data, which have a 30 m spatial resolution covering 29 scenes in Jiangxi and 31 scenes in Guangdong, were mainly obtained from the Geospatial Data Cloud website (http://www.gscloud.cn/). Data such as elevation, slope, and slope direction were calculated from the DEM data [29]. Soil erosion is one of the major factors related to soil degradation worldwide [36–38]. The widely used soil loss equation model [39] was used to calculate soil erosion in the DRB. These data were resampled to 30 m in ArcGIS.

Climate factors included average annual precipitation, average annual temperature, and relative humidity. These data were mainly obtained from the China Meteorological Data website (http://data.cma.cn/), which catalogues 17 meteorological stations. The inverse distance weighting method is a reliable method for spatial distribution that takes full account of the geographical links between factors. The above data were obtained by interpolation with the inverse distance weight method in ArcGIS 10.2.

Landscape factors included the mean patch area, boundary density, Shannon diversity index, Shannon evenness index, and Simpson diversity index, from Landsat remote sensing satellite data from the USGS website. Landsat-8 maintained a basic consistency with Landsat 1–7 in terms of spatial resolution and spectral characteristics. The satellite has a total of 11 bands, i.e., bands 1–7, and 9 bands have a spatial resolution of 30 m; band 8 is a panchromatic band with a 15 m resolution; and bands 10 and 11 have a spatial resolution of 100 m; the satellite can achieve global coverage once every 16 days [40]. For the 12 scenes, Landsat satellite image data containing few clouds were selected and are detailed in Appendix A Table A1. The calculation of landscape pattern indices, such as mean patch area [41], boundary density [41], Shannon diversity index [42], Shannon evenness index [42], and Simpson diversity index [42], was performed in Fragstats 4.2 as described at https://www.umass.edu/landeco/research/fragstats/documents/fragstats.help.4.2.pdf.

Vegetation factors, which comprise vegetative cover, were derived from Landsat remote sensing satellite data, as described in the previous subsection. The vegetative cover data were calculated using the linear spectral mixture model (LSMM) from Landsat remote sensing satellite data [43].

Socio-economic statistics included population density and GDP per capita. Population density data were sourced from the WorldPOP dataset, and GDP per capita data were sourced from the 'Statistical Yearbook of Jiangxi Province' [44] and the 'Statistical Yearbook of Guangdong Province' [45]. Population density and GDP per capita were selected to calculate the impact of socio-economic activities on the EV of the watershed. The above data were obtained using the inverse distance weighting method in ArcGIS 10.2.

*2.3. The Principal Component Analysis*

The main objective of this study was to establish a comprehensive factor system based on PCA theory for assessing EV in DRB. PCA is a method for converting existing variables into a small number of summary factors that can best represent the most original

information [46]. We first calculate the contribution rate, then look at the value of the main factor contribution rate to determine how many main factors there are, and the cumulative contribution rate of the main factors is considered to be satisfactory at 90% or more [47]. This assessment method includes a series of processing steps, e.g., the selection of relevant factors, standardization, the determination of the variance of common factors, the calculation of weights, and finally EV assessment.

### 2.3.1. The PCA Structure of EV Based on the PSR (Ecological Pressure Ecological Sensitivity, Ecological Resilience)

EV is the combined result of PSR [25,29,48,49]. Both natural and social factors can constrain the development of EV. Therefore, EV can also affect the economic development of a region, bringing about a series of developmental problems such as poor economic activity, slow development of modern industries, and low per capita education [13]. By reviewing the relevant information on EV in the DRB and assessing the relevant factor systems selected by researchers for the southern hilly mountains [50–55], it was found that a causal relationship between man and nature can be established with the PSR model. The general aim of this study was to establish a comprehensive PSR evaluation system based on PCA theory, and our analysis and evaluation system contains a target level, a criterion level, and a factor level. Both human activities and natural conditions affect EV, and so the evaluation factor system should include these two factors [56,57]. We developed a system analysis model consisting of three levels and 15 factors (Table A2 in the Appendix A). We consulted relevant experts from the Guangdong Dongjiang River Basin Authority to understand the current situation in the basin.

Ecological response factors are the most direct characteristics expressed by the long-term interactions of various factors within an ecosystem [29]. In the vicinity of the DRB, the topography is fragmented, it has varying heights of terrain, there are hilly mountains in the center and north, and deltas, lowlands, and coastal plains in the south; the DRB has a subtropical monsoon climate, with an average annual temperature of 21 °C and an average annual precipitation of 1750 mm, which is uneven and mainly concentrated in April–September [34]. Ecological sensitivity factors can be divided into topographic and climatic factors: topographic factors are selected for elevation, slope, slope orientation, and soil erosion, which directly contribute to increased soil erosion problems [58]; climatic factors include mean annual precipitation, average annual temperature, and relative humidity. The DRB is cloudy and rainy. The average annual temperature and precipitation are key climatic factors affecting the amount of vegetative production. Moreover, relative humidity can reflect vegetation transpiration, so it is crucial for ecological protection [59].

Ecological state factors have a role in protecting ecosystems and can be divided into landscape factors and vegetation factors. The mean patch area (mean patch area, area_mn) can reflect landscape heterogeneity. It can, on the one hand, constrain the minimum patches of the image landscape and, on the other hand, reflect the degree of fragmentation of the landscape [41]. The boundary density (edge density, ED) is an important factor for analyzing patch shape, as it indicates the extent to which the landscape is fragmented, i.e., the higher the value, the more the boundary is fragmented, the more dispersed the layout, and the more compact the patches [41]. The Shannon diversity index (SHDI) reflects the landscape heterogeneity, primarily used to assess the contribution of rare patches to information, i.e., the richer the land use, the greater the fragmentation in the landscape, the higher the value, and the less damage to the landscape [42]. The Shannon evenness index (SHEI) indicates the maximum likelihood of a given landscape, i.e., the richer the landscape, the healthier and more stable its ecosystem [42]. The Simpson diversity index reflects the heterogeneity of the community, and the higher its value, the more stable the ecosystem [34].

Ecological pressure factors are mainly associated with human activities that constrain the health of the ecosystem to a certain extent. The DRB is located in a humid zone. Its watershed is important to Guangdong, Hong Kong, and Macau, which include the more economically active cities of Shenzhen, Guangzhou, Dongguan, Huizhou, and Hong

Kong, the core city of the Greater Bay Area. In recent years, ecological and environmental pollution and degradation have posed serious challenges as the population has increased, and urbanization and industrial upgrading have accelerated [33]. Socio-economic activities constantly influence the magnitude of EV within the watershed area; therefore, human activities have a huge impact on the evolution of the ecological environment [60,61]. Population density increase can lead to excessive resource consumption, and increases in GDP per capita mean that more resources need to be consumed to achieve economic productivity, which constrains the health of regional ecosystems to some extent [62]. It follows that many anthropogenic activities can directly lead to environmental pollution and other serious consequences, such as the degradation and depletion of natural resources.

### 2.3.2. Weight Calculation Based on the PCA

Given that the scale criteria and inter-factor attribute interval values are different for each factor, standardization is used to maintain the same scale and criteria across factors in order to facilitate the subsequent modeling analysis [25]. All the assessment factors were controlled to be within [0, 1], i.e., values closer to 1 representing stronger vulnerability and values closer to 0 denoting weaker vulnerability. Increasing positive factors brings about increased vulnerability, and consequently, ecological damage becomes more frequent and/or even more serious. Increasing negative factors partly alleviates EV and improves the ecological environment. The formulae for the positive and negative correlation factors calculated by normalizing the above data are as follows:

$$\text{Positive: } m_i^{'} = (m_i - m_{imin})/(m_{imax} - m_{imin}) \tag{1}$$

$$\text{Negative: } m_i^{'} = 1 - (m_i - m_{imin})/(m_{imax} - m_{imin}) \tag{2}$$

where $m_i^{'}$ is the value of the first factor after standard processing; $m_i$ is the initial value of the $i$-th factor; $m_{imin}$ is the minimum of the $i$-th factor in the layer; and $m_{imax}$ indicates the maximum of the $i$-th factor in the layer.

The weights were calculated using PCA [56]. Furthermore, the weight of each principal component $\alpha_i$ was calculated using the following formula:

$$\alpha_i = \lambda_i / \sum_{i=1}^{m} \lambda_i \tag{3}$$

where $\lambda_i$ denotes the variation degree of the $i$-th principal component. The PCA results were obtained using ArcGIS10.2, and the common factor variance of each evaluation factor was calculated from the factor matrix of the results.

$$H_j = \sum_{j=1}^{m} \lambda_{jk}^2 \ (j = 1, 2, \ldots, 9; \ k = 1, 2, \ldots, m) \tag{4}$$

where $j$ is the number of identified evaluation factor factors, $k$ is the number of principal components, and $m$ is the total number of principal components. Equations (3) and (4) were used to standardize the $H_j$ (common factor variances) of the evaluation factors. Finally, the weights of each factor were obtained using Formula (5).

$$W_j = H_j / \sum_{j=1}^{15} H_j \ (j = 1, 2, \ldots, 15) \tag{5}$$

where $W_j$ denotes the weights of the $j$-th factor.

### 2.3.3. Ecological Vulnerability Model Calculation

The original evaluation factors were standardized using the extreme difference method, as was described in the previous subsection. We calculated the weighting coefficients of the 15 evaluation factors in the DRB. Moreover, we were able to construct a comprehensive EV evaluation model, which included the state of natural resources and socio-economic development in the DRB. Formula (6) is able to represent the EV of the DRB through

the comprehensive analysis of multiple evaluation factors, and thus the values of the comprehensive evaluation factors were derived to calculate the EVI of the DRB.

$$EVI_i = \sum_{i=1}^{15} p_i * W_i \tag{6}$$

where $EVI_i$ is the EV index of the *i*-th image raster, between [0, 1]; $p_i$ is the value obtained after normalization of the *i*-th factor of the raster image; and $W_i$ is the weighting factor of a factor for EV. The weighting values for the 3 years are shown in Table A2.

### 2.3.4. Threshold Definition Based on Net Primary Production

NPP (net primary production), which indicates the total amount of organic dry matter produced by green plants per unit time and unit area, can be used to measure ecological changes [10]. We used GEE (Google Earth Engine) to download annual NPP data with a pixel resolution of 500 m (m) from MOD17A3H V6 (2001, 2008, and 2019). These were introduced to assist in determining the EV thresholds for different classes within each time series.

Current studies generally define EV thresholds randomly [18,24,63]. We used the method for classifying EV described by Guo Bing [10] et al. To avoid randomness in the definition of vulnerability (EV) thresholds, we introduced NPP data to assist in the determination of EV thresholds for the 3 years (2001, 2008, and 2019), thus to some extent ensuring EV comparability in the same region over time. The main steps were as follows: (1) we divided the three NPP data periods (2001, 2008, 2019) into four classes using the equidistance method; (2) we combined the three NPP data periods to derive the EV values of the corresponding periods; (3) the EV values were reverse-ordered to obtain the EV thresholds of different classes in the three periods. The level was set at five levels: potential, slight, mild, moderate, and severe (Table 2). The potential level indicates a stable, fully functional ecosystem with a high sensitivity to external disturbance, a high self-recovery rate, and abundant vegetative cover. Slight-level ecosystems are relatively stable, fully functional, with a low sensitivity to external disturbances, a weak self-recovery rate, and good vegetative cover. The mild level indicates an ecosystem that is marginally sensitive to external disturbances but generally stable. Moderate-level ecosystems are relatively unstable, functionally deficient, find it difficult to recover from damage, are highly sensitive to external disturbances, and have poor vegetative cover. Severe-level ecosystems are extremely unstable, severely degraded, find it extremely difficult or even impossible to recover from damage, are highly sensitive to external disturbance, and have poor vegetative cover.

**Table 2.** Table of EV levels in the DRB in different periods.

| Vulnerability Level | 2001 | 2008 | 2019 | NPP |
|---|---|---|---|---|
| Potential | <0.47 | <0.41 | <0.38 | - |
| Slight | 0.47–0.51 | 0.41–0.48 | 0.38–0.42 | 0.75 |
| Mild | 0.51–0.52 | 0.48–0.50 | 0.42–0.44 | 0.5 |
| Moderate | 0.52–0.59 | 0.50–0.53 | 0.44–0.46 | 0.25 |
| Severe | >0.59 | >0.53 | >0.46 | - |

### 2.4. Geodetector

Geodetector is a statistical method for detecting spatial differentiation and revealing the factors that influence it [64]. The four detectors of the Geodetector are factor detection, interaction detection, risk zone detection, and ecological detection. The spatial distribution of EV varies significantly and is influenced by a combination of factors. We use the Geodetector to perform factor detection and interaction detection, i.e., to calculate the drivers of variation in the spatial extent of EV affecting DRB, and then to infer the interaction between two variables.

(1) Factor detector: It detects the spatial heterogeneity of EV change Y and the explanatory power of different factors X on EV change Y. Measured by the q-value, the expression is [65]:

$$q = 1 - \sum_{h=1}^{L} N_h \sigma_h^2 / N\sigma^2 \qquad (7)$$

where $h = 1, \ldots, L$, $L$ is the stratification of variable Y or factor X, i.e., classification or zoning; $N_h$ and $N$ are the number of cells in stratum $h$ and the whole area, respectively; $\sigma_h^2$ and $\sigma^2$ are the variance of stratum $h$ and the whole area of Y values, respectively. q has a range of [0, 1], with larger values of q indicating a stronger explanation of changes in EV Y by the independent variable X and vice versa.

(2) Interaction detector: analyzes the possible causal relationships between different influencing factors, i.e., whether the combined effect of different factors enhances the explanatory power of EV. In the evaluation process, we first calculate the q-values of Y for each of the two factors: q(X1) and q(X2); calculate the q-value of Y when the two layers are tangent: q(X1∩X2); and compare q(X1), q(X2), and q(X1∩X2). The relationship is detailed in the Table 3 [64].

**Table 3.** Interaction relationship.

| Criterion | Interaction |
|---|---|
| q(X1∩X2) < Min(q(X1), q X2)) | Non-linear weakening |
| Min(q(X1), q(X2)) < q(X1∩X2) < Max(q(X1), q(X2)) | Single-factor non-linear attenuation |
| q(X1∩X2) > Max(q(X1), q(X2)) | Two-factor enhancement |
| q(X1∩X2) = q(X1) + q(X2) | Independent |
| q(X1∩X2) > q(X1) + q(X2) | Non-linear enhancement |

## 3. Results

### 3.1. Temporal Evolution Characteristics of Ecological Vulnerability

On the basis of the establishment of the PSR EV model in the previous section, the spatial distribution of the EV index in the study area was calculated using Equation (6). The grading map (Figure 3) and the percentage of area in EV levels (Table 4) were calculated.

This research reclassified the EV values in 2001, 2008, and 2019 in the DRB, and then obtained three periods of vulnerability level results in this area, which are shown in Figure 4. According to the level results, the research also counted the coverage and area proportion of each classification, which are shown in Table 5. In 2001, 2008, and 2019, the EV index of the DRB ranged from 0.15 to 0.82, with an annual average of 0.51, as shown in Figure 5. According to the analysis of statistics in Table 6, the proportion of severe (V) level in the DBR was the highest in all three periods, increasing and then decreasing, indicating that EV in the DBR deteriorated and then improved. The proportion of moderate (IV) level was the lowest in all periods. The proportion of mild (III) level decreased year by year. The proportion of slight (II) level increased and then decreased, and the proportion of potential (I) level followed the opposite trend. At the macro level, the overall fragility of the eco-environment in the DRB decreased from north to south.

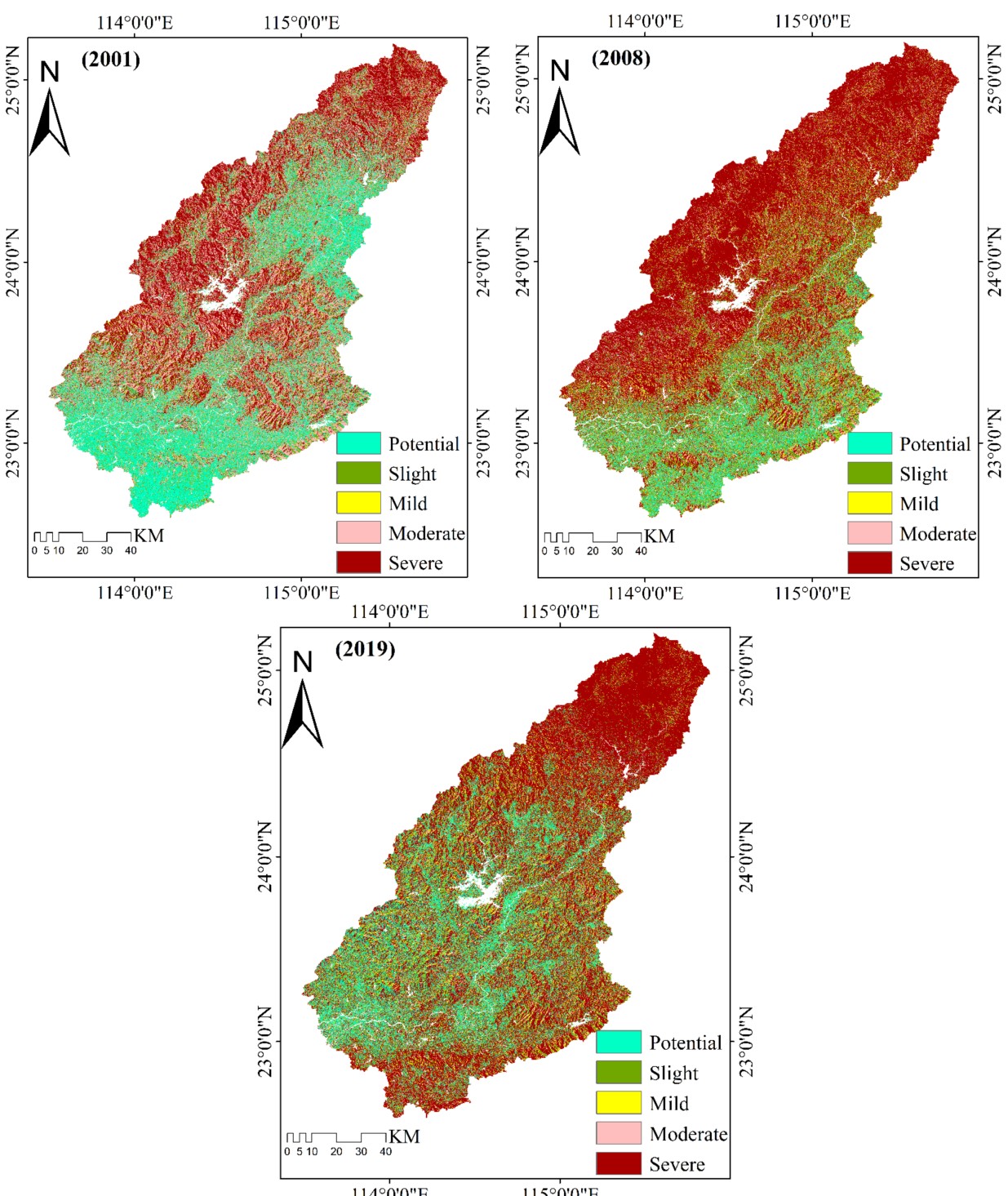

**Figure 3.** Distribution of EV levels in the DRB, 2001–2019.

**Table 4.** Statistics on the percentage of area at each EV level in the DRB (2001, 2008, 2019).

| Level | Vulnerability Level | 2001 | 2008 | 2019 |
|---|---|---|---|---|
| | | Percentage of the Total Area (%) | Percentage of the Total Area (%) | Percentage of the Total Area (%) |
| I | Potential | 23 | 8 | 17 |
| II | Slight | 14 | 17 | 15 |
| III | Mild | 4 | 7 | 9 |
| IV | Moderate | 29 | 13 | 9 |
| V | Severe | 30 | 55 | 50 |

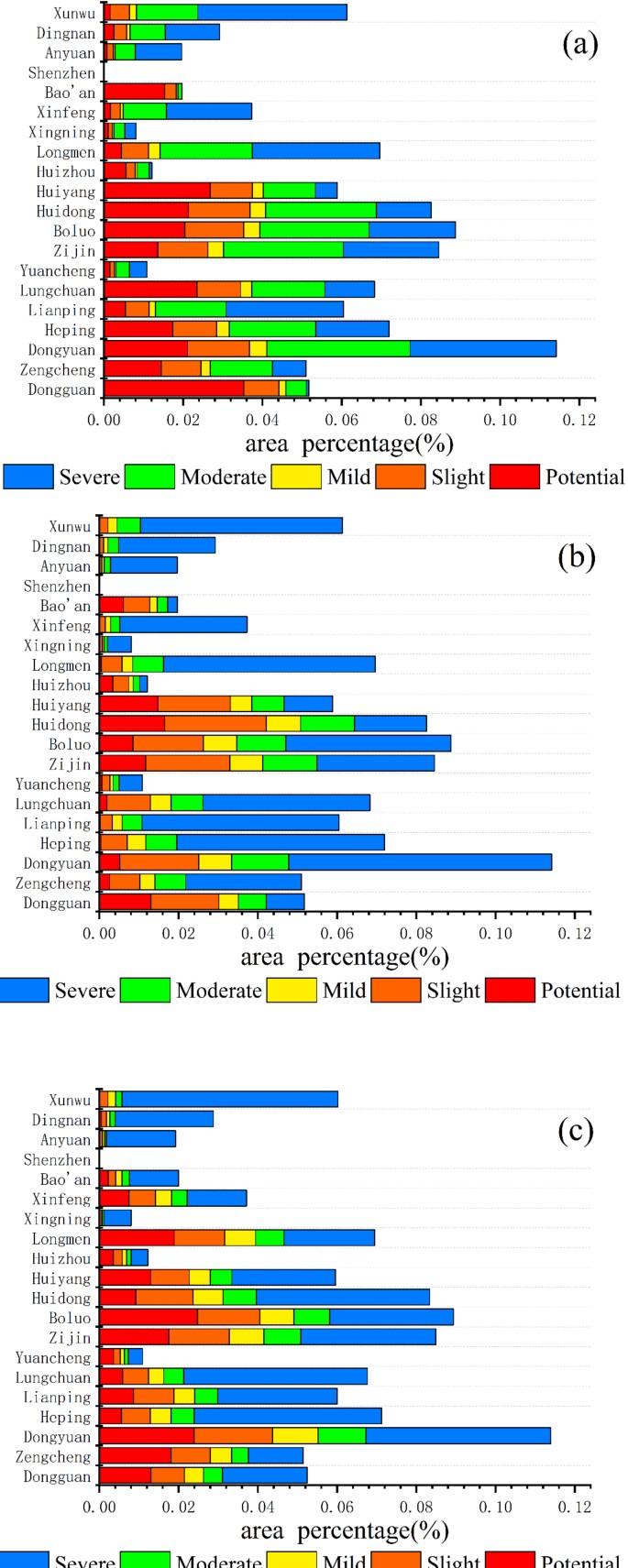

**Figure 4.** Scale map of vulnerability areas at different levels (**a**) 2001; (**b**) 2008; (**c**) 2019.

**Table 5.** Statistics of q-values for ecological vulnerability factor detection.

| Name of the Factor | 2001 | | | 2008 | | | 2019 | | |
|---|---|---|---|---|---|---|---|---|---|
| | q | q Ranking | p | q | q Ranking | p | q | q Ranking | p |
| Soil erosion(X4) | 0.109 | 10 | 0 | 0.066 | 10 | 0 | 0.037 | 12 | 0 |
| Area_mn(X8) | 0.057 | 15 | 0 | 0.046 | 14 | 0 | 0.080 | 9 | 0 |
| Slope orientation(X3) | 0.250 | 5 | 0 | 0.259 | 2 | 0 | 0.002 | 15 | 0 |
| Vegetation cover(X13) | 0.280 | 3 | 0 | 0.208 | 4 | 0 | 0.102 | 4 | 0 |
| Elevation(X1) | 0.258 | 4 | 0 | 0.207 | 5 | 0 | 0.049 | 11 | 0 |
| Boundary density (ed) (X9) | 0.057 | 14 | 0 | 0.049 | 13 | 0 | 0.084 | 5 | 0 |
| GDP per capita(X15) | 0.132 | 9 | 0 | 0.089 | 9 | 0 | 0.148 | 3 | 0 |
| Average annual precipitation(X5) | 0.245 | 6 | 0 | 0.145 | 6 | 0 | 0.076 | 10 | 0 |
| Population density(X14) | 0.172 | 7 | 0 | 0.133 | 7 | 0 | 0.023 | 13 | 0 |
| Average annual temperature(X6) | 0.284 | 2 | 0 | 0.226 | 3 | 0 | 0.235 | 1 | 0 |
| Shannon Diversity Index (SHDI) (X10) | 0.059 | 11 | 0 | 0.050 | 11 | 0 | 0.083 | 8 | 0 |
| Shannon's evenness index (SHEI) (X11) | 0.058 | 12 | 0 | 0.009 | 15 | 0 | 0.083 | 7 | 0 |
| Relative Humidity(X7) | 0.325 | 1 | 0 | 0.329 | 1 | 0 | 0.209 | 2 | 0 |
| Simpson diversity index (SIDI) (X12) | 0.058 | 13 | 0 | 0.050 | 12 | 0 | 0.084 | 6 | 0 |
| Slope(X2) | 0.144 | 8 | 0 | 0.090 | 8 | 0 | 0.015 | 14 | 0 |

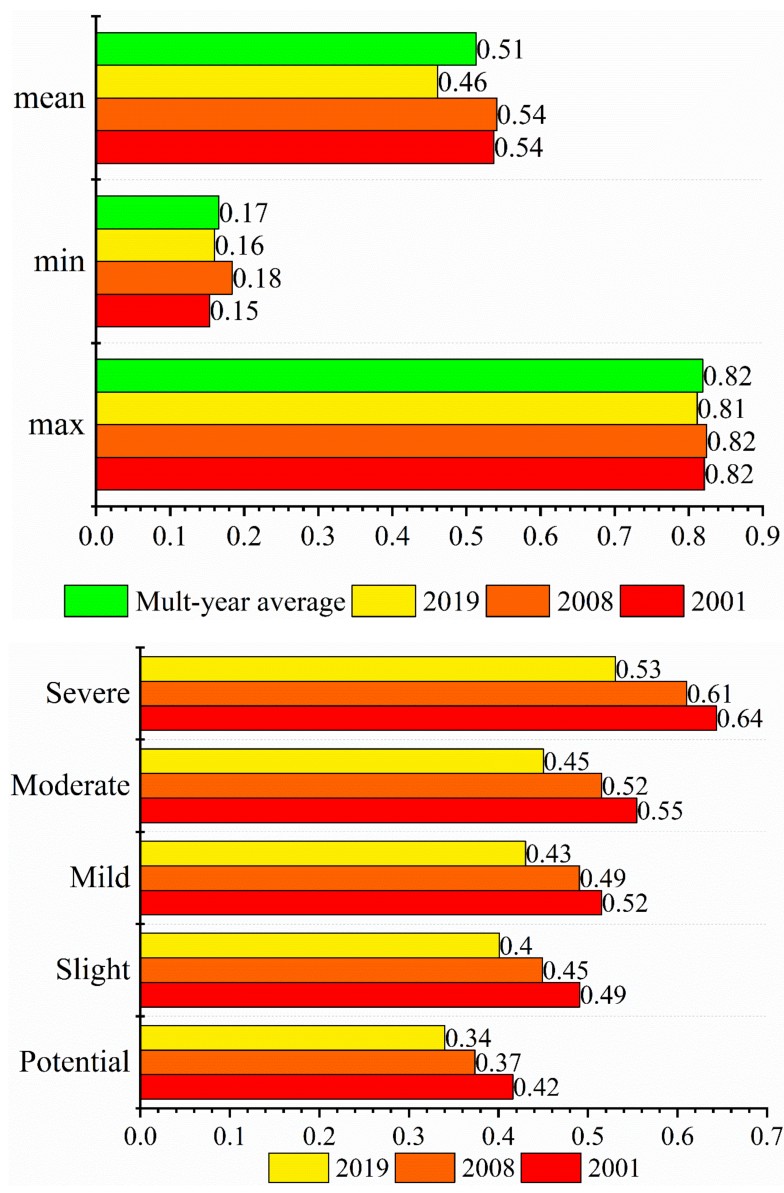

**Figure 5.** Histogram of minimum, maximum, and mean values based on EV in DRB.

**Table 6.** Interaction results for selected factors of ecological vulnerability in the DRB.

| 2001 | | 2008 | | 2019 | |
|---|---|---|---|---|---|
| X3/X7 * | 0.564 | X3/X7 * | 0.570 | X13/X6 * | 0.351 |
| X3/X6 ** | 0.535 | X13/X7 * | 0.505 | X13/X7 * | 0.321 |
| X3/X13 * | 0.528 | X3/X6 ** | 0.476 | X9/X6 * | 0.311 |
| X13/X7 * | 0.517 | X3/X13 * | 0.471 | X6/X12 * | 0.311 |
| X3/X1 * | 0.506 | X3/X1 * | 0.464 | X6/X11 * | 0.310 |
| X3/X5 ** | 0.502 | X4/X7 ** | 0.442 | X6/X10 * | 0.310 |
| X13/X6 * | 0.481 | X13/X6 * | 0.434 | X8/X6 * | 0.307 |
| X13/X5 * | 0.454 | X3/X5 ** | 0.402 | X7/X12 * | 0.291 |
| X4/X7 ** | 0.447 | X15/X7 * | 0.394 | X11/X7 * | 0.291 |
| X3/X14 ** | 0.426 | X3/X14 ** | 0.394 | X9/X7 * | 0.291 |
| X3/X2 ** | 0.405 | X10/X7 * | 0.394 | X10/X7 * | 0.290 |
| X4/X6 ** | 0.403 | X7/X12 * | 0.394 | X8/X7 * | 0.287 |
| X1/X7 * | 0.397 | X9/X7 * | 0.393 | X4/X6 ** | 0.284 |
| X5/X7 * | 0.396 | X8/X7 * | 0.388 | X5/X6 * | 0.280 |
| X1/X6 * | 0.393 | X13/X5 * | 0.379 | X5/X7 * | 0.280 |
| X15/X7 * | 0.388 | X5/X7 * | 0.379 | X13/X6 * | 0.270 |
| X6/X7 * | 0.385 | X1/X7 * | 0.368 | X15/X6 * | 0.265 |
| X3/X15 ** | 0.383 | X6/X7 * | 0.367 | X4/X7 ** | 0.257 |
| X13/X15 * | 0.382 | X15/X6 * | 0.354 | X15/X7 * | 0.255 |
| X10/X7 * | 0.380 | X3/X15 ** | 0.354 | X14/X6 * | 0.253 |

(* represents a two-factor enhanced interaction, ** represents a non-linear enhanced interaction).

### 3.2. Change of Ecological Vulnerability Grade Index

We calculated the spatial distribution of the EV change intensity in 2001, 2008, and 2019. Then, we differentiated the vulnerability maps for the three years (2001, 2008, and 2019), and the spatio-temporal changes in EV in the DRB were further analyzed. On the basis of the histogram distribution and related spatio-temporal map changes, the four stages of EV change intensity (CI) were classified as follows: decreasing intensity (CI $\leq$ −0.20); mildly decreasing (−0.20 < CI $\leq$ −0.1); stable (−0.1 < CI $\leq$ 0); mildly increasing (0< CI $\leq$ 0.1); and intensity increase (CI > 0.1). We used the ArcGIS zonal statistics tool to calculate the average of each factor for each district and county and combined it with Figure 6. The results show that, from 2001 to 2008 (Figure 6A), the DRB was dominated by two intensities, i.e., mild increase and stable, with mild increase occupying more than half and stable mainly distributed in Huidong, Zijin, Huizhou, Yuancheng, Dongyuan, and Boluo. From 2008 to 2019 (Figure 6B), mild decrease was mainly distributed in Zengcheng, Boluo, Longmen, Lianping, Dongyuan, and Yuancheng, etc., and mild increase dominated in Shenzhen, Dongguan, Huiyang, Huizhou, and Huidong in the southern DRB. The quality of the environment has improved over the last 20 years in the DRB, with a trend towards increased EV in a few areas, these being more obvious is the southernmost part of the DRB (Figure 6C).

### 3.3. Analysis of the Drivering Factors of Ecological Vulnerability

We used the ArcGIS 10.2 software to create 30 m × 30 m grid points and extracted 18,249 sample points. We relied on the aforementioned sample points to extract the values of each dependent and independent variable for the quantitative analysis and evaluation of the driving force factors. EV was selected as the dependent variable, and the 15 EV factors were selected as the independent variables. After standardizing them, the factors for each period (2001, 2008, and 2019) were transformed from numerical quantities to typological quantities using the natural breakpoint method at five levels. Finally, we used Geodetector to analyze the impact of each factor on EV. Factor detectors were used to calculate the influence of each factor on EV. As can be seen from Table 5, the *p*-value of each factor was 0, which indicates that the explanatory power of the three period factors on EV was sufficient. This also proves that the selection of topography, climate, vegetation, landscape, and economic factors to calculate the EV of the DRB was feasible.

In general, the influence of the DRB factors on EV was basically stable. From these, the top two q-values were relative humidity and average annual temperature, with mean q-values of 0.288 and 0.248, respectively, indicating that the EV of the DRB was most influenced by relative humidity and average annual temperature. This is mainly due to the fact that relative humidity and average annual temperature are the dominant factors affecting evapotranspiration in most parts of China [65]. The DRB is located in the humid subtropical monsoon climate of southern China, with abundant but unevenly distributed rainfall; the average annual temperature is 20–22 °C, with little average annual temperature variation [66]. Evapotranspiration affects regional climate change and vegetative growth, and thus the EV of the DRB [67].

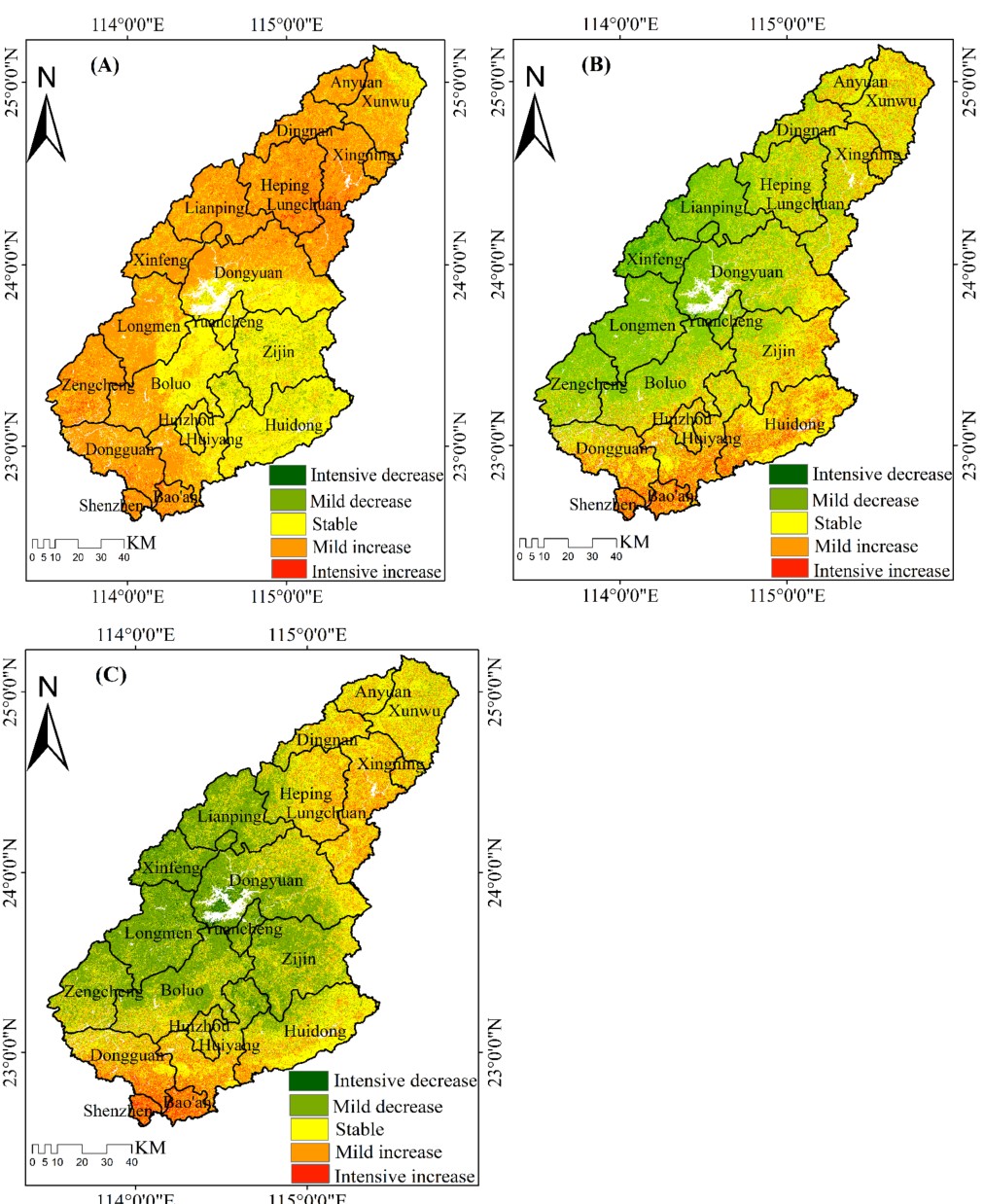

**Figure 6.** Dynamic changes in EV in DRB. (**A**) 2008–2001; (**B**) 2019–2008; (**C**) 2019–2001.

From 2001 to 2008, the q-values in the 3–6 range were mainly vegetative cover, elevation, and precipitation. They exhibit a decreasing trend, indicating that the influence of these three factors on the EV was gradually weakening. This is mainly because the DRB is located in the humid region of southern China, with undulating terrain and dense vegetation; changes in the ecological environment are largely influenced by vegetative

cover, elevation, and precipitation. The q-values in 2019 were mainly for GDP per capita, which is very different from the rankings in 2001 and 2008. As the DRB has undergone a rapid developmental process in recent years, in general, the q-value of GDP per capita exhibits an upward trend, which indicates a gradual rise in the impact of social activities on EV.

Interaction detection was used to assess whether the explanatory power of EV was enhanced when the two factors acted together. Overall (Table 6, Figure 7), the values of the EV factor interactions were greater than the maximum values of the individual factors, thus indicating that the effects of the factors on EV were not independent of each other, but occurred synergistically. As shown in the table, from 2001 to 2008, all the factors were bi-factorially enhanced, except for slope orientation, which was non-linearly enhanced with precipitation, average annual temperature, population density, and GDP per capita. In 2019, soil erosion was non-linearly enhanced with average annual temperature and relative humidity, and each was bi-factorially enhanced. From 2001 to 2019, the DRB interaction detection results were generally more stable, with relative humidity exhibiting the highest interaction with other factors. Moreover, the interaction of the relative humidity with the other factors was much higher than the number of interactions between two of the other factors that occurred. The interaction between topography, climate, vegetation, landscape, and economic factors was stronger than the interaction within each element, and the interaction between natural and social factors was stronger than the interaction within each factor, indicating that the EV of the DRB was the result of the combined effect of all factors.

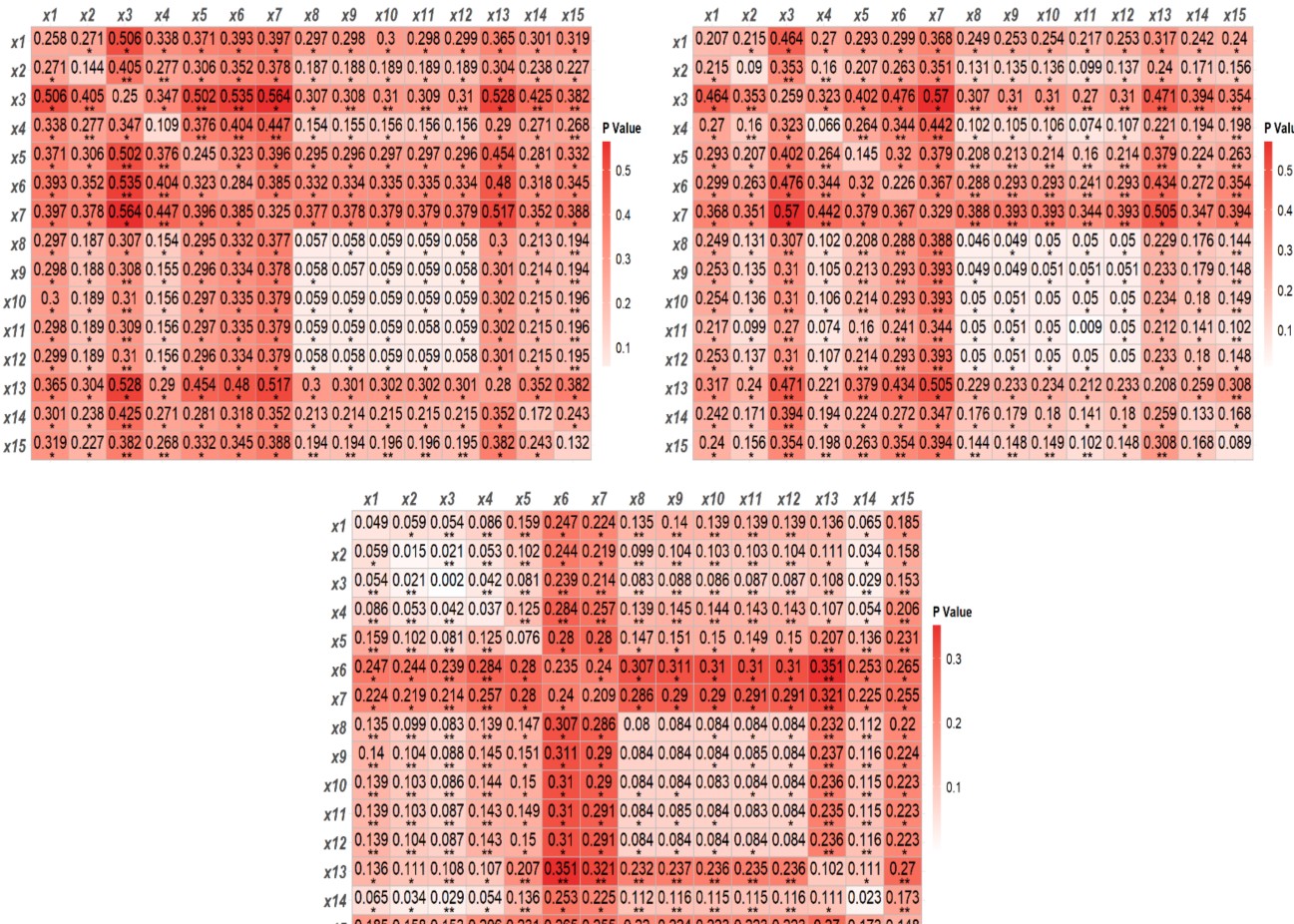

**Figure 7.** The explanatory power of interaction between factors in 2001, 2008, 2019. (* represents a two-factor enhanced interaction, ** represents a non-linear enhanced interaction, *p* value represents the interaction detection value of the two factors).

## 4. Conclusions and Discussions

*4.1. Discussions*

We took the DRB as the study area and constructed factor models to analyze the current situation through regional ecological problems. On the basis of the statistical data, such as multi-temporal remote sensing data and multi-year meteorological station data, this study used socio-economic statistics to characterize anthropogenic disturbance factors and analyze the characteristics of EV. As a result of the high level of human activity in the DRB, its ecosystem is under a certain amount of development pressure. In this study, we also conducted a scientific analysis of the manifestation factors of EV from 2001–2019 in terms of ecological pressure, ecological sensitivity, and ecological resilience, and screened out various practical and typical integrated factors. In addition, vegetation inversion, soil erosion factors, landscape pattern factors, and spatial interpolation were constructed as evaluation factors based on three time periods from 2001–2019. Finally, the DRB EV evaluation index system was established.

PCA generates principal components that are independent of each other after the transformation of the original factors, which can reduce the workload of factor selection. The weights of the principal components, i.e., the contribution rate, contain the proportion of the information from the original data to the total information, and thus, this is considered an objective and reasonable assessment. Therefore, we used PCA to evaluate the EV of the DRB. In addition, using the interpolation method, the overall trend state index, and the comprehensive EV index, with the method of dynamically determining the EV rating of NPP data used by Guo Bing [10], we finally analyzed the change in the EV rating in the area, the overall EV trend, and the quality state of each district in the DRB for three periods of vulnerability from 2001 to 2019.

The factors influencing the EV in the DRB were analyzed with the help of the Geodetector software developed by Wang Jinsong [64] et al. in order to obtain objective and scientific results. This can also be confirmed by the research results of Guo Zeqing [56] et al. and Liu Jiaru [68] et al. In this study, Geodetector was used to analyze the magnitude of interactions between various factors in the EV model for each period and the anthropogenic drivers of EV in the DRB.

On the basis of a PSR model with 15 evaluation factors, we analyzed the spatial distribution of EV in the DRB over three periods and, using a geographic probe, carried out a quantitative statistical analysis of the factors driving EV. The results show that EV is decreasing from 2001 to 2019, with a stronger explanatory power for relative humidity, mean annual temperature, and vegetation cover as the main driving factors of EV in the DRB in the 3 years. In 2019, the explanatory power of GDP per capita increases, showing that human activities further influence EV in the DRB. The relative humidity and mean annual temperature did not change much [66]. Moreover, the woodland cover of the DRB increased from 2001 to 2019 (Figure 8), showing that woodland ecosystems are the most functional and structurally complex natural ecosystems of terrestrial ecosystems. Thus, they have a decisive influence on the terrestrial ecosystem environment and play an important role in safeguarding the ecosystem [69,70]. They are not only capable of providing production and living resources for humans, they also have a variety of ecological service functions, such as water containment, carbon sequestration, oxygen release, and maintaining regional ecological balance, which are important for maintaining the environment and promoting a sustainable urban development [71]. This coincides with a consistent decrease in EV. Furthermore, in the interaction detection results, the study found that the interaction between two driving factors had a greater impact on EV in the DRB as compared to a single driving factor (Table 5, Table 6). In both driver indicators, the slope orientation (X3)/relative humidity (X7), slope orientation (X3)/average annual temperature (X6), and slope orientation (X3)/vegetation cover (X13) interactions had the strongest impact on EV in the DBR, in 2001; In 2008, the strongest interaction factors were slope orientation (X3)/relative humidity (X7), vegetation cover (X13)/relative humidity (X7), and slope orientation (X3)/average annual temperature (X6). The interaction factors of

vegetation cover (X13)/average annual temperature (X6), vegetation cover (X13)/relative humidity (X7), and (X9)/average annual temperature (X6) were strongest in 2019 (Table 6). Ecosystems are complex systems where some factors interact with each other. For example, the transpiration of the vegetation increases relative humidity, and changes in vegetation cover affect relative humidity [59]. Jiang [72] showed that vegetation has the ability to change the climate of an ecosystem. Moreover, we found that vegetation cover had a strong EV explanatory power in all three years (Table 5). Limpid waters and lush mountains are invaluable assets. Therefore, we believe that planned afforestation is a worthwhile project that may influence (improve) EV.

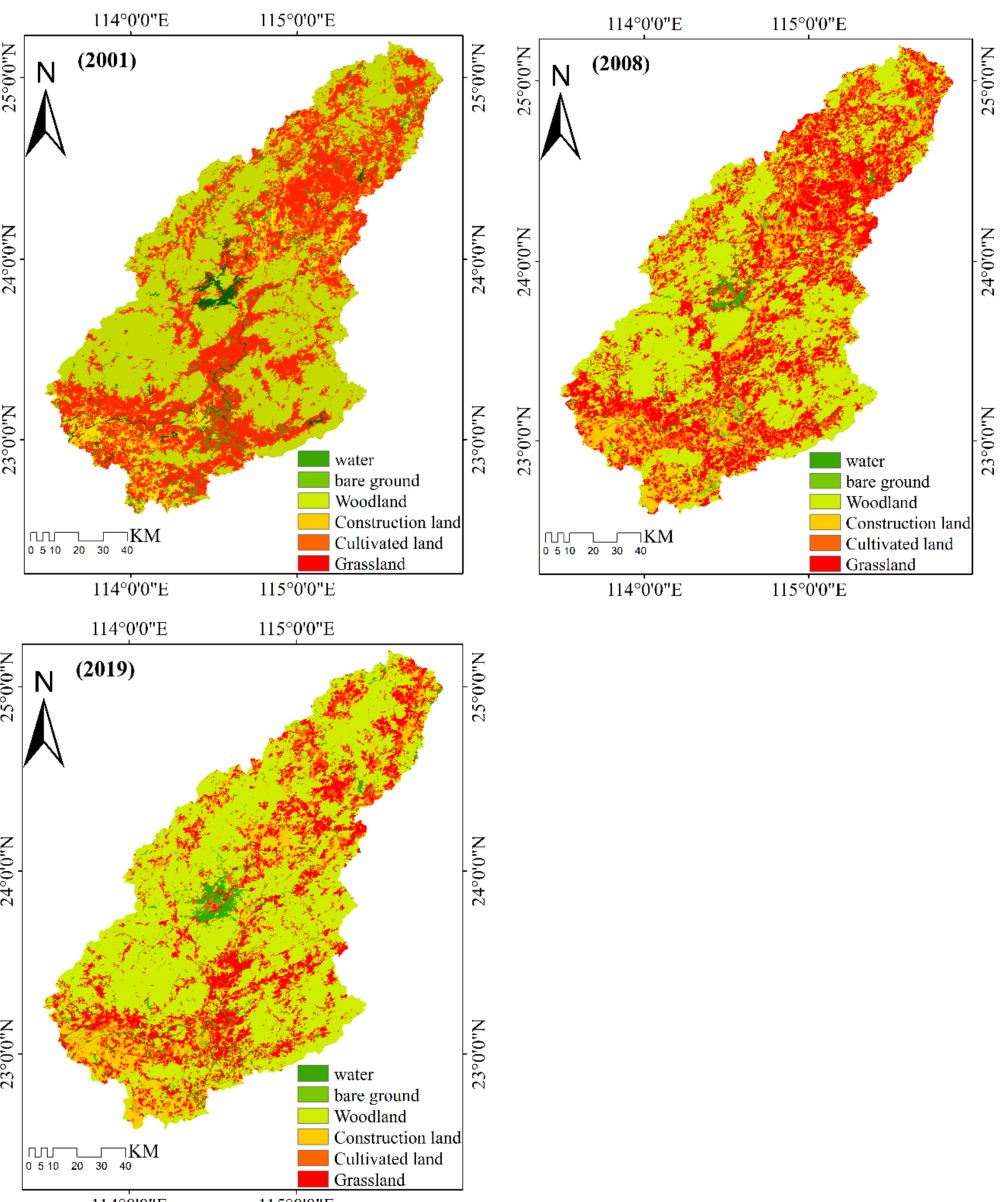

**Figure 8.** Land use map of DRB (2001, 2008, 2019).

However, the evolution of the spatio-temporal patterns of EV in the DRB from 2001 to 2019 was also influenced by many other uncontrollable natural factors and multi-level human socio-economic factors. Importantly, future studies need to cover a wide range of scientific fields and thus analyze the evolution of EV in the DRB in all its complexity. To this end, more factors must be considered in future studies.

*4.2. Conclusions*

The DRB, which is an ecologically fragile area in South China, has an important strategic position in terms of resources, environment, and ecology. In view of the unique geographical characteristics and the limitations of relevant studies, we selected 15 well-structured and statistically significant factors, such as topography, climate, landscape, and socio-economy, to analyze the EV status of the DRB. Furthermore, we analyzed its spatio-temporal changes based on the PCA analysis, and detected the driving factors using geographic probes. The following conclusions were reached:

(1) The PCA method can objectively and reasonably calculate the changes in each factor in the process of assigning weights to vulnerability factors in multi-temporal studies. The method can better reflect the change process of each factor in the EV system, and has good applicability in the southern red soil hilly ecosystem of China.

(2) NPP data can be associated with the assessment of the health of land surface ecosystems, and the EV level thresholds in different periods can be obtained with the aid of NPP data calculation, which is important for the analysis of EV in different years.

(3) Over the past 20 years, the overall EV intensity in the DRB can be characterized by a mild decrease, while the upstream and downstream EV intensity in the DRB can be characterized by a mild increase. The midstream exhibited a mild decrease. From 2001 to 2019, the mean EV value gradually decreased. From 2001 to 2008, The area of EV intensity for mild increase is much larger than stable. From 2008 to 2019, EV intensity is more widely distributed in areas of mild decrease than mild increase.

(4) During 2001–2019, the spatio-temporal pattern of EV in the DRB was significantly affected by the relative humidity, average annual temperature, and vegetation cover.

**Author Contributions:** J.W. designed experiments; J.W. and G.M. wrote the manuscripts; Z.Z., Q.H. and J.W. processed the experimental data; Z.Z. and J.W. did the methodology; G.M. supervised the manuscript; J.W., Z.Z., Q.H. and G.M. did reviewed and edited the manuscript; all authors contributed to the interpretation of the results and the writing of the paper. All authors have read and agreed to the published version of the manuscript.

**Funding:** The State Key Research and Development Plan (2018YFB10046) and This research was financially supported by China Geological Survey Project (DD20191016).

**Institutional Review Board Statement:** Not applicable.

**Informed Consent Statement:** Not applicable.

**Data Availability Statement:** The data that support the findings of this study are available from the author upon reasonable request.

**Conflicts of Interest:** The authors declare no conflict of interest.

## Appendix A

**Table A1.** Landsat image identifier, acquisition time, and type characteristics.

| Image Identifier | Acquisition Time | Sensor Type | Track Number (ROW/PATH) | Sun Elevation/(°) | Solar Azimuth/(°) |
|---|---|---|---|---|---|
| LT05_L1TP_121043_20011121_20161209_01_T1 | 21/11/2001 | | 43/121 | 39.428 | 149.338 |
| LT05_L1TP_121044_20011121_20161209_01_T1 | 21/11/2001 | TM | 44/122 | 40.544 | 148.454 |
| LT05_L1TP_122043_20011230_20161209_01_T1 | 30/12/2001 | | 43/121 | 34.59 | 147.199 |
| LT05_L1TP_122044_20081201_20161028_01_T1 | 30/12/2001 | | 44/122 | 35.663 | 146.415 |
| LT05_L1TP_121043_20081210_20161028_01_T1 | 10/12/2008 | | 43/121 | 36.556 | 150.666 |
| LT05_L1TP_121044_20081210_20161028_01_T1 | 10/12/2008 | TM | 44/122 | 37.69 | 149.878 |
| LT05_L1TP_122043_20081201_20161028_01_T1 | 1/12/2008 | | 43/121 | 37.903 | 150.895 |
| LT05_L1TP_122044_20081201_20161028_01_T1 | 1/12/2008 | | 44/122 | 39.041 | 150.076 |
| LC08_L1TP_121043_20191123_20191203_01_T1 | 23/11/2019 | | 43/121 | 41.304 | 155.234 |
| LC08_L1TP_121044_20191107_20191115_01_T1 | 7/11/2019 | OLI | 44/122 | 46.471 | 152.692 |
| LC08_L1TP_122043_20191114_20191202_01_T1 | 14/11/2019 | | 43/121 | 43.436 | 154.605 |
| LC08_L1TP_122044_20191114_20191202_01_T1 | 14/11/2019 | | 44/122 | 44.634 | 153.701 |

**Table A2.** Data sources and evaluation indicators.

| Target Layer | Criterion Layer | Indicator Layer | Name of Data | Positive and Negative | Weight of 2001 | Weight of 2008 | Weight of 2019 |
|---|---|---|---|---|---|---|---|
| Ecological Response | Terrain indicators | Elevation(X1) | GDEMV2 elevation data | Negative | 0.03598 | 0.01263 | 0.00885 |
| | | Slope(X2) | | Positive | 0.0185 | 0.00718 | 0.00407 |
| | | Slope orientation(X3) | | Positive | 0.24991 | 0.24989 | 0.2499 |
| | | soil erosion(X4) | 1:4 million Chinese soil type data | Positive | 0.00021 | 0.00017 | 0.00008 |
| | | Average annual precipitation(X5) | | Negative | 0.08956 | 0.09075 | 0.20449 |
| | | Average annual temperature(X6) | Meteorological Data | Negative | 0.04795 | 0.05347 | 0.10973 |
| | | Relative Humidity(X7) | | Positive | 0.17487 | 0.20386 | 0.11149 |
| Ecological State | Landscape indicators | Mean Patch area (Area_mn) (X8) | | Negative | 0.00059 | 0.00003 | 0.00012 |
| | | Boundary density (ed) (X9) | Landsat remote sensing satellite data | Positive | 0.00107 | 0.00042 | 0.00071 |
| | | Shannon Diversity Index (SHDI) (X10) | | Positive | 0.00059 | 0.00039 | 0.00058 |
| | | Shannon's evenness index (SHEI) (X11) | | Negative | 0.00001 | 0.00003 | 0.00012 |
| | | Simpson diversity index (SIDI) (X12) | | Negative | 0.00001 | 0.00003 | 0.00012 |
| | Vegetation indicators | Vegetation cover(X13) | | Negative | 0.24466 | 0.24056 | 0.24606 |
| Ecological Pressure | Social indicators | Population density(X14) | Population density | Positive | 0.00015 | 0.00009 | 0.00011 |
| | | GDP per capita(X15) | GDP density | Positive | 0.13595 | 0.1405 | 0.06357 |

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
