# Peer review of "Spatio-Temporal Analysis of Ecological Vulnerability and Driving Factor Analysis in the Dongjiang River Basin, China, in the Recent 20 Years"

_remotesensing, doi:10.3390/rs13224636_

Round 1

Reviewer 1 Report

Although the efforts spent by the authors to improve the manuscript, it still suffers for some weakness.

At first the general quality of the manuscript is poor, both in the figures and in the text. English language needs to be deeply revised and meany sentences needs to be re-written.

Then some points have remained unclear and without explanation: in particular the manuscript totally lacks in a real discussion of the result.

Is it consistent finding that EV has been so strongly and diffusively decreasing the the area between 2001 and 2019? You should discuss this result and compare with the "on the ground" situation.

Then, the methods explanation suffers for language weakness.

A already noticed in the previous version, from the methodological point of view, data used for EV assessment are quite different in resolutions and the final resolution is controlled by the coarser one. Then, the final resolution is ruled by the Soil data, which is 1:4.000.000 scale. Is it proper for the area in study and to discriminate among the administrative districts? You have to discuss and clarify this aspect.

Other points may be find in the annotated version of the manuscript.

Author Response

Dear reviewer,

Re: Manuscript and Title: remotesensing-1310217 andSpatiol-temporal analysis of ecological vulnerability and driving factor analysis in the Dongjiang River Basin, China in the recent 20 years

Thank you for your letter and for the reviewers’ comments concerning our manuscript (ID: remotesensing-1310217). Those comments are all valuable and very helpful for revising and improving our paper, as well as the important guiding significance to our researches. We have studied comments carefully and have made correction which we hope meet with approval. English all got mdpi language editing and the certificate will be placed in the attachment. All images have been modified and the legend font has been made larger. Revised portion are marked highlighted in yellow in the paper. The main correction in the paper and the responds to the reviewers’ comments are as following:

Reviewer 2 Report

I suggest to edit the title, since the paper does not address driving forces of the ecological vulnerability mere its indicators. In general, drivers are phenomena that trigger processes that lead to certain state. In this sense, the study does not address driving but it analyses EV factors.

Half of the abstract descripts the results of the analysis, it is too long and unnecessary.

If the scope of the paper is to “develop a reliable methodology for assessing EV” it should provide some recommendation in the conclusions, which indicators to apply and how…

Methods:

2.2. lines 144-155 It is not clear what was the basis for landscape indicators calculations – was it land cover classes derived from Landsat data? What classes?  What is difference between this land cover(?) dataset and vegetation cover derived from the same underpinning Landsat data (lines 156-159)? Why to use two similar/same datasets in the EV assessment?

How were statistical data (GDP per capita) distributed within the spatial datasets? Moreover, what was the final resolution for GIS analyses if several resolution were applied (as in table 1)?

Results:

I suggest using larger font for all map legends.

Can you explain why the Dongguan province is more than 60% severe in Fig 4 (b-2008) but not in Fig 3 (2008)?

Conclusions and Discussions:

Can you explain why is EV is decreasing from 2001 to 2019? In other words, the local ecological stability or resilience is increasing. Does it correspond to the local land use changes and global climate change? In this respect, it could help to show land cover/land use maps of the study area - in  section 2.1.

The applied EV concept consists of a mixture of several quite different factors. Could you explain the roles or influence of “stable” natural factors such as terrain (slope/orientation/elevation) and soils, and dependent factors such as land cover or vegetation cover and independent factors such as temperature and precipitation  (and derived humidity) plus socioeconomic factors such as GDP or population density? How can all this be interpreted and concluded for potential users. In your conclusions you stated that the main “drivers” of EV in DRB are relative humidity and average annual temperature, in facts, the phenomena that we cannot manage…Does it mean that we cannot influence(improve) ecological vulnerability/resilience of the landscape? That would be a very bad message…

Cited references are dominantly focused on locally based national papers and international scope is missing. Discussion must be edited thoroughly to highlight strong and weak aspects of the proposed method and its results.

The whole method in fact copies paper: Guo B, Zang W, Luo W. Spatial-temporal shifts of ecological vulnerability of Karst Mountain ecosystem-impacts of global change and anthropogenic interference[J]. Science of the Total Environment, 2020, 741: 140256. I strongly suggest to acknowledge it in the proper sections and to highlight what is its novelty or main contribution of this manuscript to the science.

Author Response

(The authors gave the same response as above.)

Reviewer 3 Report

SUMMARY

The paper addresses the research area related to the evaluation of the ecological vulnerability in the Donjiang River Basin (China) exploring the spatial-temporal pattern evolution and driving the force analysis in the past 20 years.

It aims to construct a comprehensive evaluation system of the ecological vulnerability (EV) by means of multi-source remote sensing data.

The author claim that this study can provide decision basis and technical support for ecosystem protection, ecological restoration, and management in the DRB 

BROAD COMMENTs

As a general comment, the manuscript is fluent and well structured.

MINOR COMMENTs

L143. Please consider inserting the performance reached with the IDW approach.

Figure 1. Typo error: “Landast”

Figure 2. Please consider improving the image quality

Figure 3. Please consider improving the image quality. The legends are not readable.

Figure 6. Please consider improving the image quality. The legends are not readable.

Author Response

(The authors gave the same response as above.)

Round 2

Reviewer 1 Report

The authors corrected the problems that affected the previous version of the manuscript, but some of them are still present and some new raised.

Figure 2 is still affected by some kind of stretching and the writing is to be correctly moved to the caption; is the scale bar corrected, as it differs from the previous version of the figure?

Only 7 on 17 meteorological stations are visible on the map, while all of them must be located.

Figure 3 is recalled in the text but it is not present.

Revise figure 4 x axis legend.

Figure 6 is largely different from the previous version: please explain why. Is something changed in the calculation?

In the Discussion section the sentence “strongest increase in slope orientation” change needs to be explained.

In general, more attention and cure in preparing a manuscript would be appreciated.

Author Response

Dear  reviewers,

Re: Manuscript and Title: remotesensing-1436722 and ‘Spatial–temporal analysis of the ecological vulnerability and driving factor analysis in Dongjiang River Basin, China in the recent 20 years

Thank you for your letter and for the reviewers’ comments concerning our manuscript (ID: remotesensing-1436722). Those comments are all valuable and very helpful for revising and improving our paper, as well as the important guiding significance to our researches. We have studied comments carefully and have made correction which we hope meet with approval. All images have been modified and the legend font has been made larger. Revised portions are marked highlighted in green in the paper. 

Reviewer 2 Report

Thank you for your responses and editting the manuscript.

Author Response

Dear  reviewer,

Re: Manuscript and Title: remotesensing-1436722 and ‘Spatial–temporal analysis of the ecological vulnerability and driving factor analysis in Dongjiang River Basin, China in the recent 20 years

Thank you for your letter and for the reviewers’ comments concerning our manuscript (ID: remotesensing-1436722). Those comments are all valuable and very helpful for revising and improving our paper, as well as the important guiding significance to our research. We have studied comments carefully and have made correction which we hope to meet with approval. All images have been modified and the legend font has been made larger. Revised portions are marked highlighted in green in the paper.

This manuscript is a resubmission of an earlier submission. The following is a list of the peer review reports and author responses from that submission.

Round 1

Reviewer 1 Report

The manuscript in its actual form suffers for some major weakness and errors. The authors may find both major and minor points in the annotated version of the manuscript.

In general, the first aspect is related to the not correct use of sections: informations regarding methods and data must stay only in the relative sections and not in others. This means that everything regarding all the methodological aspects, the relative coincise and clear explanation and proper citations must stay in the Methods section. As an example, only in the Discussion section the reader knows that the area is largely carsick. Section 2 must be fully restructured and then properly re-written.

What is related to the results after the methods application must stay in the Results section, and these results must be discussed in the relative section. Finally, Conclusion must not be simply a brief summary of the work.

Then another important problem is related to citations: in many parts of the manuscript there is a lack of proper citations. Please remember that if you make a statement without citations, it means that it is your statement and then you have to prove it.

From the methodological point of view, data used for EV assessment are quite different in resolutions and the final resolution is controlled by the coarser one. Then, the final resolution is ruled by the Soil data, which is 1:4.000.000 scale. Is it proper for the area in study and to discriminate among the administrative districts? You have to discuss and clarify this aspect.

Some passages in the methodology are very unclear and must be presented clearly in the methods section. As an example the approach used in the Discussion section must be clearly stated. Some difficulties and inaccuracies  are present even in the sub section 2.2 and must be corrected.

A point that is not clear is why the EV thresholds for the class division are different for the three periods? This means that classes are not consistent among the periods and then that no comparison among them is possible. Besides, after solving the previous issue and considering the spatialize data, even if affected by the resolution issue, a quantitative comparison would be necessary and would help in gaining informations about changes: for this scope you could easily use Cohen's Kappa calculation. 

Then another important point emerges at lines 298-299: bear in mind the difference between climate and meteorology factors. You state that the second ones follow a linear trend: who says that? Then you find trends in data in figures 7 and 8, but data are quite scattered and trends do not emerge clearly. Besides, it is not clear what the points stand for: are you mixing data from the various districts in the different periods? It is not clear.

The statement at line 432-434 must be explained: the increase of population density is correlated to greener and sustainable development? This phrase must be supported by data.

Please clarify where are the -a and -b curves at lines 450 451.

Finally I wish to encourage the authors to improve the manuscript revising it properly: all the notations are present in the annotated version of the manuscript, both the major and the minor. 

Reviewer 2 Report

The authors provide an analysis of Environmental Vulnerability for a region of China. This seems a reasonable goal. However these kinds of multi-criteria assessments are quite complex and require a very clear structure in the presentation so that the steps are processes are fully explained and understood by the reader. At present the structure of this manuscript does not achieve this. I started out to work my way through the manuscript and provide comments and corrections that you will see below. However eventually it became clear that the manuscript and the analysis and the presentation are deeply flawed. 

  1. Abstract: English phrasing needs improvement. e.g. line 20. "In 2008, the up-20 stream showed intensity and medium intensity vulnerability." The "intensity scale" is not explained in the abstract so these sentences do not convey much.
  2. Abstract: More precision with terminology also required. e.g. line 12-14 "Landsat remote sensing series satellite data, DEM data, meteorological data, soil type data, NPP data, 13 and socio-economic statistics." Many of these acronyms not defined here and phrasing is awkward. Just "Landsat Thematic Mapper time series" would be adequate. And there is no need to continually state "data". Just "Digital Elevation Models, meteorology, soil types, net primary production data and...".
  3. Introduction: Line 70. "new highland" what does this mean. Do you mean "level" of openness etc??
    1. Line 80-81. "but the quantitative data were small and the 81 qualitative components were large," This is an awkward phrase. Something like " but the study used limited quantitative and was heavily dependent on qualitative inputs" would be much better. 
  4. Methods: Line 110 etc. "Socio-economic activities within the basin are more developed and greatly disturb their environmental changes." This makes no sense.  Do you mean that "A significant increase in the disturbance of the natural environment has been  driven by rapid socio-economic development"?
    1. Section 2.2. This needs to be rewritten with better structure. You have to explain the conceptual basis for the assessment system first. i.e. How is EV framed? This means an initial explanation: you postulate that EV is made up of, ES, ER and EP. Next you define four top level criteria for assessing these element: terrain, landscape, vegetation and societal. These are constructed in turn from a number of layers: landform, soil and climate layers for terrain;  etc etc. At present Table 1 which is vital is very disorganized. It is impossible to understand which layers contribute to which criteria. It is also not clear what kind of processing platform is involved. is a GIS? Is it some in-house processing system, is it a spatial multi-criteria analysis package?
  5. OK. I see the problem. You have to present your framework first. That means Figure 2 has to come first. This is where you explain how you are framing the analysis. Current Section 3.1 is not in the methods. It is in a section 2 of its own (2. Conceptual Framework) and described the conceptual basis for the analysis. You have to explain first why you choose elevation, slope etc as contributors to ES. An why you choose patch stats and diversity indices to measure ER. And why you choose pop and GDP density to represent EP. This has to come first. You also need to explain why you choose to construct a set of indicators layers and then throw them into a PCA. The basis for the model is not clear. There are lots of technical elements and equations. But the concept is not clear. How does this analysis fit together? How do indicators build ES, ER and EP and then go to make up EV. The reader has to clearly understand this up front, before all the data description and various procedures for creating layers and models, etc.
  6. Tables 2 and 3 belong in an appendix or supplement. Table 2 is far too detailed and Table 3 adds nothing. You can state the three dates of imagery in the text as you have and refer to these Tables.
  7. The numbering of sections in the manuscript is wrong. Section 3 contain section 2.1.1. More care is needed.
  8. Indicators. What is an indicator? You must specifically define each of these. Vegetation coverage is not an indicator. What does it mean? In the section 2.1.1 a lot of jargon and algorithm stuff is given but at the end the reader is no wiser as to what the vegetation indicator is? Section 2.1.2. Average patch area is a metric of landscape heterogeneity. Right. You need a Table that described exactly what each indicator is and what it is meant to represent. Section 2.1.3. This is not a lottery, you don't say that terrain attributes such as x, y, z were calculated. You define the importance of specific indicators and why you need them.

I am afraid there is just too much wrong with this manuscript and this study for me to continue. I do not believe that even major revision will overcome the deficiencies in the manuscript and the science understanding.

Therefore, regretfully I must recommend rejection.

Reviewer 3 Report

OVERALL

Dear authors and Editor,

this manuscript is a case study assessing ecological vulnerability (EV) in the Dongjiang River Basin (China) in 2001, 2008 and 2019. There are several case studies on EV worldwide, and I also read two such papers published in the Science of the Total Environment (https://www.sciencedirect.com/science/article/pii/S0048969718332923; https://www.sciencedirect.com/science/article/pii/S0048969720367103). So, being another case study, this manuscript requires a huge effort in many aspects to be further considered. I focused on the ‘Materials and Methods’ and in the beginning I thought it is me that I can’t understand all the methods that the authors applied, but then I read the above mentioned papers and fully understood the methods described there. Thus, I strongly advise the authors to read these (and maybe other) previous papers on EV to help them write a comprehensive ‘Materials and Methods’ section. For example, in this one https://www.sciencedirect.com/science/article/pii/S0048969718332923, check how they detail the description of ‘standardization and weighting of factors’, (note, you say ‘factors’ and you mean ‘indicators’, you describe the indicators and then you suddenly call them factors, this confuses the reader). Note how they describe what a ‘positive indicator’ is, whereas you call it ‘positive correlation indicator’ without describing what it means. Also, note how EV classification is described, and note that it is described in the ‘Materials and Methods’, not in the ‘Results’ section, and with enough detail for the reader to understand what each EV class denotes.

Write a more specific Introduction and remove unnecessary information (e.g. about COVID-19; or explain how COVID-19 relates to the concept of EV). Start by describing the concept of EV, what is it and how is it commonly assessed, which indicators are usually used, and which aspects of ecosystems are threatened by human pressures and how the EV concept relates to these aspects. Finish your Introduction by ‘The purpose of this study was to …’ and be specific on what you did. You did not assess EV during 2001-2019; you assessed EV in 2001, 2008 and 2019 and discuss on trends etc.

I didn’t read the Discussion in detail but I noticed you say that you used SPSS and assessed EV based on four indicators?? In the Materials and Methods you say you used 15 indicators, did I miss anything? Don’t show us results in the Discussion. Just discuss the results of your study based on previous literature. Don’t describe new methods in the Discussion. Describe all methods and how many indicators you used (15 or 4?) in the Materials and Methods.

Tables and figures should be self-explained. This means that their captions need to have enough detail for the reader to understand what the tables/figures show without any help from the main text. The caption of Table 1 is just ‘Data sources and evaluation indicators’. It should be ‘Ecological vulnerability (EV) indicators (structured in three layers) and their relative weights, selected and assigned for assessing EV in the Dongjiang River Basin, China’. Please adapt all Table and Figure captions accordingly.

To conclude, I got really confused after reading this manuscript. I cannot recommend it for further consideration as t needs thorough reforming to be considered for publication, but this would result to a completely different paper. Thus, I cannot engage in a line-by-line review as the manuscript lacks consistency, needs re-structuring to fit appropriate information at the right sections, requires a new discussion because the current one is mostly new methods and results, not discussion. Once again, I strongly suggest the authors to read the two above mentioned papers and adapt their manuscript accordingly.

Kind regards,  

SOME DETAILED COMMENTS

MATERIALS AND METHODS

Lines 118-158. I assume these lines were accidentally inserted because the information they contain is repeated after 159. If not, you may merge information from these lines with relevant information from lines after 158 and start your ‘Materials and Methods’ from line 158.

Line 160. I didn’t find this sensitivity-resilience-stress framework of EV in the studies you cite [23,35-40]. A Pressure-State-Response (PSR) triplet is more often and I suggest to include your indicators in this PSR framework. However, you could apply your framework after proper description of what ecological sensitivity, resilience etc. are, as very nice detailed in this paper https://www.sciencedirect.com/science/article/pii/S0048969720367103.

Line 161. This statement is very vague and gives no information to the reader. Which natural and which social factors can constrain EV and how? Just remove such vague statements.

Lines 161-164. These are also vague statements. How can EV affect economic development and cause all these problems that you state? Besides, this is not ‘Materials and Methods’. Remove lines 160-164 and start you ‘Materials and Methods’ from line 166: ‘Fifteen indicators were selected to assess ecological vulnerability (EV) in the DRB. These were (1) elevation (m a.s.l.), (2) slope, (3) slope orientation, (4) vegetation cover (m2), (5) soil erosion, (6) average annual temperature (oC), (7) average annual precipitation (mm), (8) average annual relative humidity, (9) mean patch area, (10) boundary density, (11) Shannon’s diversity index, (12) Shannon’s evenness index, (13) Simpson’s diversity index (14), Gross Domestic Product (CNY) and (15) population density (people/km2)’.

Line 170. By ‘PSR’ do you mean ‘Pressure-State-Response’ or ‘Pressure-Sensitivity-Resilience’? Convert your framework to the more common ‘pressure-state-response’ framework.

Figure 2. Please keep a consistent use of variable/indicator names. In the text you say ‘soil erosion’, here ‘soil erodibility’. In the text ‘mean patch surface base’, here ‘average patch area’. Vegetation cover -> vegetation coverage. You are delivering a scientific document, mind your text, mind consistency. Also, see Table 1 from this paper https://www.sciencedirect.com/science/article/pii/S0048969718332923, adapt your Table 1 and inset it here to describe your 15 indicators. Again, keep consistent, in Table 1 you say ‘target level’ but you obviously mean ‘target layer’?

Table 1. So, ‘Positive and negative’… what positive and negative? Do you mean ‘Positive/negative weights’? Why is ‘Positive’ written with a capital ‘P’ and ‘target’ with a small ‘t’?

Lines 174-239. Describe how all your 15 variables were derived/calculated in the same sequence reported in lines 166-171. Start by elevation, slope etc.

Line 191. Have you previously defined LSMM?

Line 205. … conducted in FRAGSTATS 4.2 (McGarical, 2015). https://www.umass.edu/landeco/research/fragstats/documents/fragstats.help.4.2.pdf.

Lines 207-213. Maybe provide the links to the websites?

Lines 221-222. Shouldn’t you provide a citation for the EPIC model? This one maybe? https://www.jstor.org/stable/76847.

Line 228-232. What is NPP and where is NPP in the 15 indicators you calculated and show in Figure 2? What is GEE? Which indicator was calculated using NPP data?

Lines 245-246. What is ‘positive correlation indicators’ and ‘negative correlation indicators’? What kind of correlation is this? You did not mention anything about correlation previously, did you?

Line 249. Standard processing? How was an indicator standard-processed?

Lines 252-287. I did not understand how you used the PCA to assign weights to your indicators. In contrast, I fully understood how this is done by reading this paper https://www.sciencedirect.com/science/article/pii/S0048969720367103. Shouldn’t you apply the PCA to reduce your 15 indicators to a number of principal components (PC) and then assign the weight based on the contribution of each PC from the PCA? Is this what you did but it is not mentioned adequately? Please adapt your relevant description based on the above paper.